# Blood-Based DNA Methylation Biomarkers to Identify Risk and Progression of Cardiovascular Disease

**DOI:** 10.3390/ijms26052355

**Published:** 2025-03-06

**Authors:** Tarryn Willmer, Lawrence Mabasa, Jyoti Sharma, Christo J. F. Muller, Rabia Johnson

**Affiliations:** 1Biomedical Research and Innovation Platform, South African Medical Research Council, Tygerberg 7505, South Africa; lawrence.mabasa@mrc.ac.za (L.M.); jyoti.sharma@mrc.ac.za (J.S.); christo.muller@mrc.ac.za (C.J.F.M.); rabia.johnson@mrc.ac.za (R.J.); 2Centre for Cardio-metabolic Research in Africa, Division of Medical Physiology, Faculty of Medicine and Health Sciences, Stellenbosch University, Tygerberg 7505, South Africa; 3Division of Cell Biology, Department of Human Biology, Faculty of Health Sciences, University of Cape Town, Cape Town 7925, South Africa; 4Department of Biochemistry and Microbiology, University of Zululand, Kwa-Dlangezwa 3886, South Africa

**Keywords:** non-communicable diseases, cardiovascular disease, DNA methylation, global, candidate gene, genome-wide

## Abstract

Non-communicable diseases (NCDs) are the leading cause of death worldwide, with cardiovascular disease (CVD) accounting for half of all NCD-related deaths. The biological onset of CVD may occur long before the development of clinical symptoms, hence the urgent need to understand the molecular alterations underpinning CVD, which would facilitate intervention strategies to prevent or delay the onset of the disease. There is evidence to suggest that CVD develops through a complex interplay between genetic, lifestyle, and environmental factors. Epigenetic modifications, including DNA methylation, serve as proxies linking genetics and the environment to phenotypes and diseases. In the past decade, a growing list of studies has implicated DNA methylation in the early events of CVD pathogenesis. In this regard, screening for these epigenetic marks in asymptomatic individuals may assist in the early detection of CVD and serve to predict the response to therapeutic interventions. This review discusses the current literature on the relationship between blood-based DNA methylation alterations and CVD in humans. We highlight a set of differentially methylated genes that show promise as candidates for diagnostic and prognostic CVD biomarkers, which should be prioritized and replicated in future studies across additional populations. Finally, we discuss key limitations in DNA methylation studies, including genetic diversity, interpatient variability, cellular heterogeneity, study confounders, different methodological approaches used to isolate and measure DNA methylation, sample sizes, and cross-sectional study design.

## 1. Background

Cardiovascular diseases (CVDs), including coronary heart disease (CHD), hypertension, and heart failure, represent the highest prevalence of non-communicable diseases worldwide and are increasing at an alarming rate, particularly in low- and middle-income countries [1,2]. In 2019, CVDs were responsible for approximately 17.9 million deaths, representing 32% of all deaths globally [3]. CVDs are progressive disorders which involve an intricate interplay between genetic, environmental, and lifestyle factors, including tobacco use, physical inactivity, alcohol consumption, and obesity [4,5]. Additional risk factors include social adversity, high blood pressure, pollution, and urbanization in low- and middle-income countries [5,6].

Investments in prevention are considered to have a major impact on reducing the CVD burden. By preventing and/or delaying the disease, it may be possible to reduce CVD-associated mortality and the economic burden of treating this disease. Furthermore, the use of biomarkers may aid in the identification of high-risk individuals, profile key modifiable risk factors, and represent novel therapeutic targets. To this end, epigenetic alterations that are detectable in the blood, a minimally-invasive biological source, may present as attractive biomarkers and are thus drawing increasing attention for the early detection of many diseases [7].

Epigenetics, defined as heritable changes that influence gene expression without alteration(s) within the underlying DNA sequence, may shed light on the interplay between the genome and its environment in complex diseases, such as in CVDs [8]. DNA methylation is an epigenetic modification defined as the enzymatic addition of a methyl group to the fifth carbon of cytosine dinucleotides [9]. DNA methylation within the promoter regions of genes has been extensively linked with gene silencing, and accumulating studies have uncovered important roles for DNA methylation in the etiopathogenesis of CVDs [10,11,12,13,14,15]. There is also evidence to suggest that DNA methylation may be an early event in the onset of CVDs and, importantly, since DNA methylation is reversible, screening for disease-associated DNA methylation marks in asymptomatic individuals may assist in the early detection of disease and serve to predict response to therapeutic interventions in affected individuals.

For studies focused on CVDs, the accessibility and procurement of cardiac tissue, which is central to disease pathogenesis, is not feasible. This has led to a search for less invasive clinical biomarkers associated with CVDs that can be detected in easily accessible tissues, such as in peripheral blood. While some studies have reported that whole blood or its constituents are good proxies for DNA methylation in disease-associated tissues, there is still a paucity of information on whether this is the case for CVDs [16,17].

This review aims to explore the current evidence on blood-based DNA methylation signatures that are associated with CVDs. Three bibliographic databases (Cochrane, PubMed, and Scopus) were searched to identify articles reporting on CVD-associated DNA methylation signatures in blood using the following keywords: “DNA methylation”, AND “blood”, OR “peripheral blood”, OR “peripheral blood mononuclear cells”, OR “peripheral blood leukocytes”, OR “peripheral blood lymphocytes”, OR “white blood cells” AND “cardiovascular disease” AND “human”. As CVD is an umbrella term encompassing a wide range of phenotypes, this review included studies that examined both general CVD outcomes as well as related conditions affecting the heart and blood vessels. To this end, we included studies investigating myocardial infarction, coronary heart disease, stroke, heart failure, and atherosclerosis, recognizing the shared pathophysiological mechanisms and overlapping risk factors of these conditions. To maximize the sample size of the included articles, no restrictions on dates were applied to the literature search. Articles were considered eligible if they were original, full-length studies investigating the DNA methylation modifications associated with CVD or related conditions in humans, using randomized controlled trials, cohort, case-control, cross-sectional, or longitudinal study designs. Intervention studies, case reports, letters, conference abstracts, and articles not written in English were excluded. The reference lists of all eligible original research and review articles were also manually scanned to identify studies that were missed using the original search strategy.

## 2. Studies Profiling DNA Methylation in CVD

A total of 4747 articles were identified in the initial literature search. After removing duplicates, article titles and abstracts were screened for eligible full-text articles, of which 37 met the inclusion criteria. An additional 11 studies, identified from reference lists of eligible articles, were also included in this review (Figure 1). The detailed characteristics of the eligible studies investigating the DNA methylation alterations in CVDs are summarized in Table 1, Table 2 and Table 3. A total of 12 studies analyzed global DNA methylation, 14 investigated DNA methylation of candidate genes, and 22 investigated genome-wide DNA methylation signatures. The studies were conducted in numerous populations, and included European (n = 17), Chinese (n = 16), American (n = 12), Indian (n = 4), African (n = 4), Japanese (n = 1), Iranian (n = 1), Turkish (n = 1), and Brazilian (n = 1) participants. The included articles studied either cross-sectional, case-control datasets (n = 30) or longitudinal cohorts (n = 18), with the participant sample sizes varying substantially between studies (ranging from 2 to 11,273 individuals). Thirty-five of the studies included a sample larger than 100 individuals and three studies analyzed samples from 20 or fewer individuals. The profiling of DNA methylation was conducted in either whole blood or blood constituents, including white blood cells, peripheral blood mononuclear cells, lymphocytes, and leukocytes.

### 2.1. Global DNA Methylation Studies

Global DNA methylation is defined as the total percentage of cytosines within the genome that are methylated [63]. A change in global DNA methylation has been frequently associated with aging and cancer, and more recently, with metabolic diseases, including CVDs [64]. We identified several articles investigating the status of global DNA methylation in major CVD processes, including atherosclerosis, hypertension, and CHD (Table 1). These studies used various molecular techniques to quantify global DNA methylation, including the measurement of repetitive elements as a surrogate for DNA methylation, enzyme-linked immunosorbent assays (ELISAs), methylation-specific restriction enzymes, as well as liquid chromatography mass spectrometry (LCMS).

Castro and colleagues (2003) investigated global DNA methylation levels in the peripheral blood leukocytes from 17 male atherosclerosis patients compared to healthy controls, using a methylation sensitive restriction enzyme-based assay [19]. The authors reported that global DNA methylation was significantly reduced in the atherosclerosis patients and inversely correlated with plasma homocysteine levels [19]. Since homocysteine plays a prominent role in one-carbon metabolism and is an established, independent risk factor for vascular disease [65], the authors proposed that elevated plasma homocysteine concentrations observed in the atherosclerosis group may consequently result in DNA methyltransferase inhibition and reduced systemic methylation capacity. Similarly, using an ELISA method, Ramos (2016) reported reduced global DNA methylation in the peripheral blood leukocytes from 90 postmenopausal women, which revealed an association between global DNA hypomethylation and cardiovascular risk that remained significant even after adjustment for time since menopause [25].

It is estimated that more than half of the human genome is comprised of repetitive elements [66]. These include retrotransposons, such as LINE-1 and Alu elements, which have gained increased attention in the last decade for their use as surrogate markers of global DNA methylation. In line with the findings of Castro (2003) and Ramos (2016), six additional studies analyzing the status of LINE-1 and Alu in blood identified an association between global DNA hypomethylation and CVD risk predictors, such as altered levels of low density- and high-density lipoproteins, hypertension, and CVD events, such as stroke, CHD, and myocardial infarction [21,22,23,24,67,68]. Interestingly, there is also evidence to suggest that methylation of LINE-1 and Alu elements may be sensitive to environmental exposures. Indeed, numerous CVD risk factors, including exposure to air pollution [69,70], cigarette smoking [71], and alcohol consumption [72], have also been associated with DNA hypomethylation, suggesting this epigenetic signature may be an early event in the development of CVDs, as well as other lifestyle-related diseases.

Other data from global DNA methylation studies have been conflicting. Kim et al. (2010) investigated methylation of Alu elements in 286 participants from the Singapore Chinese Health Study and identified a positive correlation between DNA hypermethylation in peripheral blood leukocytes and the prevalence of CVDs [10]. Using a methylation-sensitive restriction enzyme assay, Sharma and colleagues (2008) also observed an increase in the global DNA methylation levels in the peripheral lymphocytes from 137 patients with CHD compared to 150 healthy controls, which also correlated with plasma homocysteine levels [20]. Similarly, using a pyrosequencing approach, Alexeeff et al. (2013) assessed the DNA methylation of LINE-1 and Alu elements in relation to blood pressure in a longitudinal study comprising 789 elderly participants from the Normative Aging Study, and identified a positive association between Alu hypermethylation and diastolic blood pressure, whilst no changes in LINE-1 methylation were observed [27].

Moreover, Jiang and colleagues (2019) investigated the levels of 5-methylcytosine (5mC) and 5-methylhydroxycytosine (5hmC) in a cross-sectional cohort consisting of 91 Chinese participants diagnosed with atherosclerosis and 22 healthy controls. They identified a positive correlation between 5mC/5hmC and Gensini scoring, a clinical indicator of atherosclerosis severity. Furthermore, 5mC/5hmC significantly correlated with atherosclerosis risk factors, including fasting glucose, HbA1c levels, and Crouse score [28]. In an additional study by the same group, 5mC/5hmC levels were analyzed in the peripheral blood mononuclear cells from 44 elderly subjects with CVDs (23 with acute myocardial infarction and 21 with angina) and 42 control subjects matched for age and gender. Using two independent quantification methods, i.e., ELISA and dot plot, the authors equivocally showed that 5mC levels were significantly increased in the CVD patients compared to the controls [29]. Lastly, in contrast to the aforementioned findings, Istas and coworkers (2017) investigated global DNA methylation in whole blood samples from eight atherosclerosis patients compared to eight healthy controls from the Impact of Le Ventricular Assist Devices Implantation on Micro- and Macrovascular Function cohort [LVAD, (NCT02174133)] and found no significant difference in global DNA methylation between the two groups [26].

### 2.2. Candidate Gene Studies

Despite the popularity of global DNA methylation studies, a shift towards investigating gene candidates that correlate with specific clinical features or disease traits has gained increased interest in recent years [73]. The methodological approaches used for this application include methylated DNA immunoprecipitation (MeDIP), quantitative methylation-specific PCR, mass spectrometry, and bisulfite pyrosequencing. Using these approaches, several studies have identified associations between the methylation status of specific gene loci and cardiovascular risks. These loci include *fatty acid binding protein 3* (*FABP3*), *interferon gamma* (*IFN-γ*), *interleukin-6* (*IL-6*), *inducible nitric oxide synthase* (*iNOS*), *thioredoxin-interacting protein* (*TXNIP*), *bone morphogenic protein receptor 2* (*BMPR2*), *forkhead/winged helix transcription factor 3* (*FOXP3*), *miR-510*, *miR-223*, *sterol O-acyltransferase 1* (*SOAT1*), *human telomerase reverse transcriptase* (*hTERT*), *toll-like receptor 2* (*TLR2*), *glucocorticoid receptor* (*NR3C1*) and *atrial natriuretic peptide* (*NPPA*), and are elaborated on in the subsequent sections and summarized in Table 2. 

#### 2.2.1. Genes Involved in Inflammation and Immunity

Chronic inflammation is largely mediated by the expression of interleukin genes, which encode a family of pro-inflammatory cytokines [74]. Of these, IL-6, which regulates various inflammatory responses, has been the most extensively studied. Zuo and colleagues (2016) investigated the methylation status of two CpG sites within the IL-6 promoter in the peripheral blood leukocytes obtained from 212 CHD patients [36]. Both CpG sites were hypomethylated in the CHD patients compared to the controls, with the combined methylation scores correlating significantly with CHD risk [36]. Whilst the gene and protein expression in this study were not determined, other studies have reported a positive relationship between circulating IL-6 expression and CVD characteristics, including blood pressure and arterial stiffness [75,76]. Importantly, IL-6 expression is also influenced by exposure to CVD risk factors, including air pollution and salt consumption, highlighting that aberrant DNA methylation and the expression of IL-6 may be an early and preventable event in the development of CVDs [75,77]. In an independent study, DNA methylation of the IL-6 promoter was investigated in a sample of 35 Iranian patients with atherosclerosis versus 30 healthy controls. The study identified hypomethylation of six CpG sites in the distal IL-6 promoter in the peripheral blood from atherosclerosis patients compared to healthy subjects, which inversely correlated with mRNA expression levels [39]. Interestingly, the CpG sites investigated in this study were highly conserved amongst several species, indicating that these sites may functionally be important in the regulation of IL-6 gene expression.

In a 10-year longitudinal study involving 789 elderly participants, Alexeeff and colleagues (2013) investigated the methylation status of a panel of inflammatory genes in whole blood using a pyrosequencing approach [27]. After adjusting for confounding factors, including age, body mass index, diabetes mellitus, alcohol consumption, ethnicity, and neutrophil count, they reported that DNA methylation of the *TLR2* and *iNOS* promoters positively correlated with blood pressure, whilst a negative correlation was observed between *IFN-γ* promoter methylation and blood pressure. Although the expression of these genes was not quantified in the study, *TLR2*, *IFN-γ*, and *iNOS* have previously been implicated in atherosclerosis and myocardial inflammation in both humans and animal models [78,79,80,81]. Indeed, overexpression of *TLR2* in mice leads to early atherosclerosis onset, whilst *TLR2*-deficient mice display reduced atherosclerosis-associated inflammation [82]. *IFN-γ* is a T-cell-derived, macrophage-activated pro-inflammatory cytokine that is overexpressed in human atherosclerotic lesions [83]. In mice, the exogenous administration of *IFN-γ* exacerbates the formation of atherosclerotic lesions, whilst the knockout of *IFN-γ* reduces lesion size [83]. Furthermore, rabbits exposed to a high-cholesterol diet, whilst treated simultaneously with a selective *iNOS* inhibitor, were ameliorated from the development of atherosclerosis compared to control atherosclerotic rabbits [84].

Increased expression of circulating *TXNIP*, a member of the alpha-arrestin protein family, has been associated with numerous metabolic disorders [85]. *TXNIP* is upregulated in response to high-glucose-induced stress and plays a role in redox homeostasis by binding to and activating the NOD-like receptor family pyrin domain containing three (NLRP3) inflammasome [85]. The methylation status of *TXNIP* was investigated in the peripheral blood leukocytes of 54 Chinese patients with CHD compared to 54 age- and gender-matched healthy controls. The results showed that a single CpG site, located at cg19693031 within the 3′ UTR of *TXNIP*, was hypomethylated in the CHD patients compared to the controls. The DNA methylation at this site also inversely correlated with *TXNIP* mRNA levels, as well as fasting plasma glucose and total cholesterol [42]. In line with this study, the DNA methylation of this same CpG site was found to be associated with type 2 diabetes in four independent GWAS studies involving Mexican-American, German, Indian-Asian, and Spanish populations [86,87,88,89].

Valvular heart disease is a common cause of pulmonary hypertension and is associated with high mortality rates, partly due to a lack of reliable and effective early diagnosis markers [90]. A study by Li et al. [30] investigated the methylation status of *BMPR2*, a member of the TGF-β family, in the peripheral blood from Chinese patients with valvular heart disease complicated by pulmonary hypertension (VHD-PAH). The study cohort consisted of 26 male and 29 female VHD-PAH patients and 28 controls. Using pyrosequencing, the authors showed that the *BMPR2* promoter was hypermethylated in cases versus controls, which inversely correlated with *BMPR2* mRNA levels. Importantly, receiver operating characteristic curve (ROC) analysis revealed that DNA methylation combined with *BMPR2* mRNA levels could distinguish VHD-PAH patients from controls with high specificity. These findings agree with studies showing a link between *BMPR2* mutations and the development of pulmonary arterial hypertension [91]. These findings also agree with other studies highlighting monoallelic mutations, resulting in haploinsufficiency of *BMPR2*, as the main genetic risk factor for heritable pulmonary arterial hypertension [92]. Moreover, homozygous and heterozygous mutations in *BMPR2* in rats lead to spontaneous pulmonary hypertension, as evidenced by pulmonary artery pressure, pulmonary vascular resistance, right ventricular hypertrophy, and decreased cardiac output [93].

Regulatory T (Treg) cells have been demonstrated to play a protective role against atherosclerosis in numerous experimental models. These cells are regulated by FOXP3, and recently, genetic variants of the *FOXP3* gene have been identified as risk factors for the development of atherosclerosis [94,95]. Zhu and colleagues analyzed the DNA methylation levels at a conserved regulatory region (known as the Regulatory T-Cell-Specific Demethylated Region) within the *FOXP3* locus in the peripheral blood from a longitudinal cohort consisting of 171 patients with acute CHD, and showed that elevated methylation levels of *FOXP3* associated with poor clinical outcomes and severity of atherosclerosis [31]. This agrees with an independent study showing that DNA methylation within the same regulatory region of *FOXP3* was associated with CHD, even after adjusting for age, smoking, BMI, mean blood pressure, blood glucose, and lipid profiles [32]. The authors of this study additionally showed that exposure of cultured peripheral blood mononuclear cells to oxidized low-density lipoprotein resulted in a dose-dependent increase in *FOXP3* methylation. The authors thus proposed a mechanism whereby dysregulation of Treg cells may occur via epigenetic suppression of *FOXP3*, leading to an increased risk of developing acute coronary syndrome [32].

#### 2.2.2. MicroRNAs

MicroRNAs (miRNAs) are short, non-coding regulatory molecules that control gene expression post-transcriptionally [96]. The altered expression of miRNAs has been associated with the development of a growing list of non-communicable diseases, including cancer, diabetes, and cardiovascular disease [96]. Krishnan et al. (2017) investigated the promoter DNA methylation levels of *miR-510* in the peripheral blood from 20 hypertensive patients versus 20 healthy controls using a methylation-specific PCR-based method. Their study revealed that the *miR-510* promoter was hypomethylated in hypertensive patients, which was accompanied by increased *miR-510* gene expression levels [33]. *MiR-510* has an established role in regulating cell proliferation, migration, and apoptosis, and not surprisingly, it has been implicated as an onco-miRNA in several cancers [97,98,99]. While the detailed role of miR-510 in hypertension remains to be elucidated, the authors proposed that circulating *miR-510* may be a promising candidate diagnostic biomarker for hypertension.

*MiR-223* plays a crucial role in cholesterol homeostasis by targeting genes involved in the synthesis, degradation, and transport of lipoproteins. Expectedly, it has been implicated in diseases involving the vascular system and has been proposed as a circulating biomarker for a range of cardiovascular diseases, including atherosclerosis, myocardial infarction, arrhythmia, and hypertension [100]. A study by Li and colleagues (2017) investigated the methylation status of the *miRNA-233* promoter in the peripheral blood leukocytes from 23 patients with atherosclerotic cerebral infarction compared to 32 healthy controls [37]. Using bisulfite sequencing, they identified a total of seven hypomethylated CpG sites within the *miRNA-233* promoter in cases versus controls, which negatively correlated with *miRNA-233* mRNA levels. Using Spearman’s correlation analysis, the authors showed that methylation at these CpG sites inversely correlated with total plasma cholesterol levels. The mean methylation levels of these sites were also lower in participants with carotid atherosclerosis compared to those without carotid atherosclerosis. The authors thus proposed that *miRNA-233* should be considered as a potential biomarker and target for the prevention and treatment of atherosclerotic cerebral infarction.

#### 2.2.3. Genes Involved in Lipid Metabolism

Dysregulated lipid metabolism is a common feature of obesity, insulin resistance, and CVDs [101]. Fatty acid binding proteins (FABPs) are abundantly expressed in the heart and serve as chaperones for intracellular lipids, particularly for the transportation of free long-chain fatty acids for cellular metabolism, resulting in protection against lipid toxicity [102]. A previous preclinical study in mice showed that the depletion of FABP3 results in increased glycolysis, lipid accumulation, reduction in fatty acid uptake, and cardiac dysfunction [103]. Based on this preliminary evidence, Zhang et al. (2013) sought to investigate the DNA methylation levels of *FABP3* in PMBCs, and its relationship with cardiometabolic risk factors in a study cohort consisting of 517 individuals of Northern European descent [34]. Using an Epityper mass array, they investigated the DNA methylation of 17 CpG sites located within the proximal promoter and first exon of *FABP3*. The average methylation of the 17 CpG sites correlated positively with cholesterol levels, whilst methylation at specific individual sites linked with diastolic blood pressure, insulin resistance, and low-density lipoprotein. Importantly, the gene expression of *FABP3* was found to be inversely correlated with *FABP3* methylation and positively correlated with protective metabolic factors, HDL-cholesterol, and adiponectin, and negatively with total cholesterol, triglycerides, and waist-to-hip ratio [34]. The inverse relationship between DNA methylation and *FABP3* gene expression is consistent with a previous study showing that a 50-base pair region directly upstream of the *FABP3* transcription start site was sufficient to drive expression of the *FABP3* gene in a mouse model [104].

The Sterol-O-Acyl transferase (SOAT) enzyme family consists of membrane-bound proteins localized to the endoplasmic reticulum, where they play a key role in converting endoplasmic reticulum cholesterol to cholesterol esters to store lipid droplets [105]. Of these enzymes, *SOAT1* has been linked to inflammation and the formation of atherosclerotic plaques. Moreover, depletion of *SOAT1* was previously shown to ameliorate dysfunctional insulin signaling and dyslipidemia in mice exposed to a high-fat diet [106]. Based on these studies, Abuzhalihan and colleagues (2019) investigated *SOAT1* methylation in a Chinese population of 99 individuals with CHD and 89 healthy controls. Of 26 CpG sites analyzed within the *SOAT1* promoter, 7 CpG sites had lower methylation levels in the CHD patients compared to the control group. These methylation profiles correlated with early disease onset after adjusting for total cholesterol [41].

#### 2.2.4. Genes Associated with Stress and Aging

It is well established that telomere length is associated with age-related diseases, including a broad spectrum of cardiovascular diseases [107]. Amongst various levels of regulation, telomere activity and length are maintained by hTERT. The gene encoding hTERT has been shown to be regulated by methylation in a wide range of tissue types [108]. Zhang and colleagues (2013) investigated promoter methylation of *hTERT* in 197 atherosclerosis patients and 165 age- and sex-matched healthy controls. They found that *hTERT* was demethylated in atherosclerosis patients compared to healthy individuals, which corresponded with a decrease in telomere length and *hTERT* mRNA levels [35]. Although the mRNA changes observed in this study did not match the longstanding relationship between the promoter DNA methylation and gene silencing, the authors found that hypomethylation of the *hTERT* promoter occurred within binding sites specific to CTCF, a transcriptional repressor. They therefore speculated that demethylation of the *hTERT* promoter, in this scenario, may promote the recruitment of this repressor and consequently block *hTERT* transcription.

The glucocorticoid receptor is a critical regulator of the HPA axis, a neuroendocrine pathway associated with the stress response [109]. Genetic and epigenetic alterations within the *NR3C1* gene, encoding the glucocorticoid receptor, have been implicated in neuropsychiatric diseases and more recently, a range of cardiometabolic disorders [110]. A study by Zou et al. (2015) investigated the relationship between promoter methylation of *NR3C1* and brachial artery flow-mediated dilation as a measure of subclinical atherosclerosis in the peripheral blood leukocytes from 84 monozygotic male twin pairs from the Vietnam Era Twin Registry. After adjusting for covariates (smoking, physical activity, body mass index, lipids, blood pressure, and fasting glucose) and multiple testing, they identified 12 CpG sites in the *NR3C1* exon 1F promoter that displayed intra-pair differences in DNA methylation, which positively correlated with brachial artery flow-mediated dilation. The authors speculated that altered glucocorticoid signaling, through dysregulated *NR3C1* methylation, may result in endothelial dysfunction through the disruption of the HPA axis, which influences metabolic signaling [38].

#### 2.2.5. Genes Involved in Endothelial Function

*NPPA* encodes a peptide hormone that is produced within the atria of the heart. Polymorphisms within the *NPPA* gene have been associated with CHD, atherosclerosis, and ischemic stroke, and functional studies have indicated that NPPA promotes diuresis, natriuresis, and vasodilatation, and may thus play fundamental roles in the pathogenesis of CVDs through these mechanisms [111,112]. In a prospective, longitudinal study conducted in China, 2498 middle-aged and elderly participants were followed for a total of 10 years, after which 263 participants developed CVD-related events. Using bisulfite sequencing, the authors investigated the DNA methylation status of nine CpG sites within the proximal promoter of the *NPPA* gene in peripheral blood. The results revealed that hypermethylation of a CpG site situated 459 base pairs upstream of the transcriptional start site was significantly associated with reduced CVD risk, which was independent of age, sex, education level, tobacco and alcohol use, body mass index, fasting glucose levels, and circulating lipids [111,112]. In this regard, the methylation status of *NPPA*, as proposed by the authors, may be useful in predicting the risk of lifestyle diseases, including CVDs.

### 2.3. Genome-Wide Studies

The advent of next-generation sequencing has led to a major paradigm shift in epigenomic research, facilitating a hypothesis-free assessment of DNA methylation signatures at a genome-wide scale [113]. Almost a decade ago, a study published by Lister et al. (2009) described, for the first time, the use of Whole-Genome Bisulfite Sequencing (WGBS) to generate comprehensive DNA methylation maps of the entire human genome at single-base pair sites [114]. Now, whole-genome sequencing is routinely utilized to identify genes and genomic regions that are differentially methylated in relation to disease onset and progression. These sequencing platforms include Infinium Beadchip Arrays (HumanMethylation27, HumanMethylation450, and HumanMethylationEPIC arrays), sequencing of immunoprecipitated methylated DNA, and whole-genome bisulfite sequencing [115]. This section will provide an overview of genome-wide, differentially methylated CpG regions associated with cardiovascular outcomes, which is summarized in Table 3. Of the 22 studies included 1 study utilized a methylation specific restriction-enzyme based microarray, 2 studies conducted methyl-binding domain-capture sequencing (using the Illumina HISEq 2500), 2 studies carried out reduced representation bisulfite sequencing (RRBS), while 4 studies used the HumanMethylationEPIC BeadChip array, and 13 studies used the older HumanMethylation450 BeadChip array.

Using a CpG island microarray approach, Sharma et al. (2014) identified hypermethylation at 19 CpG regions in the peripheral blood mononuclear cells from male patients with coronary artery disease (CAD) compared to controls [43]. The putative biological function(s) of the differentially methylated genes were further investigated by gene ontology analysis, which revealed an enrichment of genes involved in signal transduction, transcriptional regulation, organelle development, transport, and cell regulation. Of the 19 differentially methylated regions identified, 12 selected regions were subjected to bisulfite sequencing using samples from an independent, validation cohort comprising 48 CAD cases and 48 control individuals. The results showed hypermethylation of six CpG sites mapping to intronic regions within the *STE20-related kinase adaptor alpha* (*STRADA*) gene and *Heat Shock Protein 90 Beta Family Member 3 Pseudogene* (*HSP90B3P*), as well as exon 1 of the *Complement C1q Like 4* (*C1QL4*) gene. While there is a paucity of information on the roles of *STRADA* and *HSP90B3P* in CVD development, *C1QL4* has previously been implicated in cardiovascular system development through its ability to promote the angiogenesis of endothelial cells [116].

Using the HumanMethylation450 platform and two different population cohorts, Guarrera and colleagues (2015) investigated DNA methylation profiles in white blood cells and their association with myocardial infarction occurrence [24]. They identified a differentially methylated region within the gene body of *Zinc Finger and BTB Domain Containing 12* (*ZBTB12*), which was hypermethylated in myocardial infarction cases compared to controls. Furthermore, the association of this differentially methylated mark was more pronounced in subjects with a shorter time to disease, revealing its potential as a prognostic marker. A study by Rask-Andersen and colleagues (2016) performed genome-wide DNA methylation analysis on blood samples from a cross-sectional study population cohort from Northern Sweden. This cohort included individuals with hypertension, myocardial infarction, stroke, thrombosis, and cardiac arrhythmia. They found that 211 CpG sites, representing 196 genes, were differentially methylated in individuals with a history of myocardial infarction. Gene ontology analysis revealed an enrichment of genes involved in neuronal systems, transmembrane ion transport, and synaptic signaling. Furthermore, a large proportion of differentially methylated loci (n = 42) mapped to genes previously associated with cardiac function and CVDs in the literature, such as *Dysferlin* (*DYSF*) and *Neuregulin 1* (*NRG1*), supporting the biological relevance and candidacy of these methylation marks as potential biomarkers for myocardial infarction [44].

Armstrong (2018) studied the DNA methylation patterns of 180 individuals (aged 35 years and older) in a multicenter, double-blinded, randomized controlled clinical study involving the Vitamin Intervention for Stroke Prevention cohort. The study population comprised Europeans (n = 104) and African Americans (n = 76), all of whom had suffered recurrent episodes of ischemic stroke within 120 days of enrolment and displayed high plasma homocysteine levels. Data from this study revealed two highly significant hypermethylated CpG sites, mapping to the *ASB10* (cg22812874) and *TTC37* (cg00340919) genes, which were associated with an increased risk of stroke. Of particular interest, this association was found exclusively in the European subgroup but was absent in the African American subgroup [45]. Moreover, since the European patients suffered recurrent stroke episodes, it is still unknown whether the DNA methylation changes observed in *ASB10* and *TTC37* were causal or a consequence of ischemic stroke.

Age is an established and independent cardiovascular risk factor that autonomously alters epigenetic signatures. In a study conducted by Fernández-Sanlés (2018), the author investigated the influence of age on DNA methylation in individuals with atherosclerotic traits using whole blood from two independent longitudinal cohorts, the Registre Gironí del Cor (REGICOR) cohort and the Framingham Offspring Study (FOS) cohort [46]. Using the Illumina HumanMethylation (450K) BeadChip array, they identified 48 differentially methylated regions associated with cardiovascular risk in the REGICOR study. These signatures were further validated in the FOS cohort, and findings revealed the hypomethylation of six CpG loci within four genes, namely *aryl hydrocarbon receptor repressor* (*AHRR*, cg05575921), *carnitine palmitoyltransferase 1A* (*CPTIA*, cg00574958), *peptidylprolyl isomerase F* (*PPIF*, cg12547807, cg19939077, cg18608055), and *strawberry notch homolog 2* (*SBNO2*, cg18608055), which correlated with increased cardiovascular risk after adjusting for age. Importantly, post-analysis showed that the DNA methylation of these CpG loci was able to explain 8.53% and 15.6% of the CVD risk in the REGICOR and FOS cohorts, respectively. This study also identified an association between hypomethylation of these CpG loci and several CVD risk factors, including smoking (*AHRR* and *PPIF*), BMI (*PPIF*), and the combination of BMI, diabetes, and triglyceride levels (*CPTIA*). BMI additionally correlated inversely with the methylation of a single CpG site mapping to the *Growth/Differentiation Factor-15* (*GDF-15*) gene, an established CVD biomarker with well-reported roles in inflammation. Indeed, *GDF-15* has been shown to be overexpressed in the myocardium and blood of patients with acute myocardial infarction [117,118] and its levels are associated with increased risk of death in individuals with CHD and chronic heart failure [119].

Bain and colleagues (2020) performed a pilot investigation of the DNA methylation differences in the peripheral blood from 20 male CAD patients with extreme heart failure phenotypes compared to 20 age-, sex-, and ethnicity-matched CAD patients without these extreme phenotypes. Using a methyl-binding domain-capture sequencing approach, the authors identified 68 differentially methylated regions associated with extreme heart failure. Gene enrichment analysis identified 103 significantly enriched gene sets, of which hypomethylated CpGs annotated to *histone deacetylase 9* (*HDAC9*), *jumonji and AT-rich interaction domain containing 2* (*JARID2*), and *gremlin 1* (*GREM1*), whilst hypermethylated CpGs annotated to *decaprenyl diphosphate synthase subunit 2* (*PDSS2*) [47]. An additional study by Gao (2021), followed by Ammous (2022), made use of the Infinium HumanMethylationEPIC Beadchip 850K array to identify novel methylated biomarkers that can predict the risk of developing CVDs in Chinese and African American populations, respectively. In the study by Gao (2021), eight CpGs were differentially methylated, with one hypermethylated CpG [annotating to *cytochrome P450 family 8 subfamily B member 1* (*CYP8B1*, cg07655795)], being associated with increased CVD risk, whilst hypomethylation of the remaining seven CpGs [annotating to *spondin 1* (*SPON1*, g06901278, cg11651314), *phosphofurin acidic cluster sorting protein 1* (*PACS1*, cg03914662), *uveal autoantigen with coiled-coil domains and ankyrin repeats* (*UACA*, cg09306458, cg05359217), and *coiled-coil domain containing 50* (*CCDC50*, cg05946546, cg02518222)], associated with an increased risk of developing CVDs [48]. Furthermore, Ammous et al. (2022) identified a significant correlation between hypomethylation at four CpGs, annotating to *aryl hydrocarbon receptor repressor* (*AHRR*, cg05575921 and cg21161138), *TIMP metallopeptidase inhibitor 3* (*GF11*, cg09936388), and *leucine rich repeat containing 52* (*LRRC52*, cg181168448), and atherosclerosis risk in an African American population. Interestingly, two of these identified CpGs (cg05575921 and cg21161138, annotating to *AHRR*,) were also similarly associated with CVDs in two independent studies by Fernández-Sanlés (2018, 2021) in American and European populations [46,50].

A growing list of epigenome-wide studies have been used to identify DNA methylation changes associated with serum lipids, which remain the most established risk factors for CVDs. Hedman (2017) proposed that CHD-associated DNA methylation changes may not necessarily be causal to CHD, but a consequence of previous insults on the myocardium, such as lipotoxicity. To test this, the authors conducted an epigenome-wide association study using the Illumina HumanMethylation450K BeadChip array to identify the DNA methylation alterations that are associated with lipid concentrations, which may serve as potential predictors of CHD events [51]. Findings from the discovery study identified 193 CpGs in the two independent population cohorts, of which 33 could be validated in an additional 2025 study participants. Of these, 25 CpG sites showed novel associations with lipids, including those annotated to the *ATP binding cassette subfamily G member 1 locus* (*ABCG1*, cg27243685), a gene associated with reverse cholesterol transport, which subsequently correlated with a 38% higher risk of incident CHD per every standard deviation increase in methylation [51]. Interestingly, this gene has also previously been identified as aberrantly methylated in the peripheral blood from type 2 diabetic individuals compared to healthy individuals [64].

To determine the association between DNA methylation and body mass index (BMI), Mendelson et al. (2017) utilized whole blood-derived DNA samples collected from patients enrolled in the Framingham Heart Study (FHS) and Lothian Birth Cohorts (LBCs), and validated their findings using samples collected from patients in the Atherosclerosis Risk in Communities (ARICs), Genetics of Lipid Lowering Drugs and Diet Network (GOLDN), and Prospective Investigation of the Vasculature in Uppsala Seniors (PIVUSs) studies [52]. The study reported a three-way association between 11 CpG sites, the expression of seven genes, and BMI. Of the identified CpGs, two annotated to *CPT1A* (cg00574958 and cg17058475), one annotated to *ABCG1* (cg01881899), while one annotated to *sterol regulatory element binding transcription factor 1* (*SREBF1*, cg11024682), with the latter shown to associate with increased adiposity and CAD. The methylation of the cg11024682 site in the *SREBF1* gene has previously been associated with lipid measures in other studies, suggesting that the role of SREBF1 in lipid metabolism, in the context of cardiometabolic risk, may be alleviated by modulation of its methylation profile [120].

A study by Allum (2019) investigated the association between cardiometabolic risk-linked epigenetic loci and circulating plasma lipid levels in matched adipose tissue and whole blood samples from participants with obesity from the Quebec Heart and Lung Institute (IUCPQ) study. Using methylC-capture-sequencing, they analyzed 1.3 million CpG sites mapping to adipose tissue-specific regulatory regions. After adjusting for sequencing depth, age, and BMI, the authors identified 1230 CpG sites associated with at least one plasma lipid trait, i.e., triglycerides, HDL-cholesterol, LDL-cholesterol, and total cholesterol [53]. These lipid-associated CpG sites were annotated to 264 putative adipose tissue-specific enhancers and 303 promoters, of which 341 were shared elements between other tissues and adipose tissues, and 226 were specific to adipose tissue. These differentially methylated regulatory regions annotated to several obesity-related loci, including *tyrosine-protein kinase* (*CSK*), *solute carrier organic anion transporter family member 3A1* (*SLCO3A1*), *G protein subunit gamma 7* (*GNG7*), *G protein subunit alpha i1* (*GNAI5*), *lipocalin 2* (*LCN2*), *enoyl-CoA hydratase short chain 1* (*ECHS1*), *isocitrate dehydrogenase 2* (*IDH2*), and *CD7 molecule* (*CD7*). Next, they aimed to determine whether these signatures could be replicated in matched whole blood from the same sample population. Out of the 565 regulatory regions identified in the adipose tissue, 340 were reflected in the whole blood, of which 68 (mapping to 13 putative enhancers and 55 promoters) shared the same associations with plasma lipid traits, including those annotating to *serine/threonine kinase AKT1*, *histone deacetylase HDAC4*, *bone morphogenetic protein 4* (*BMP4)*, *growth differentiation factor 7 (GDF7)*, *ceramide kinase* (*CERK*), *vestigial like family member 3* (*VGLL3*), and ATP*-binding cassette transporter 5* (*ABCC5*). Of the 68 CpG sites identified in the IUCPQ cohort, 22 could be further validated in the whole blood of an independent cohort. Taken together, these findings offer distinct characteristics of both tissue-specific and tissue-independent regulatory regions associated with lipid profiles and cardiometabolic risk [53].

C-reactive protein (CRP), an established marker of heightened inflammation, has been associated with CVDs and stroke. In a recent study by Chilunga (2021), the impact of DNA methylation loci, specifically those linked to CRP, on the development of CVDs was explored in a population of 589 African individuals with type 2 diabetes. The study highlighted 14 previously unidentified differentially methylated loci associated with CRP levels reaching up to 40 mg/L, with three of the differentially methylated CpGs [annotating to *pyruvate carboxylase* (*PC*, cg14653250), *BTG anti-proliferation factor 4* (*BTG4*, cg13767940), and *peptidyl arginine deiminase 1* (*PAD1*, cg21010178)] demonstrating a link with CVD risk [54]. While *BTG4* and *PADI1* have not yet been linked to CVD-related events, the *PC* gene has been shown to play an important role in gluconeogenesis, lipogenesis, and insulin secretion and has recently been associated with T2D [121].

In an attempt to assess the blood-based DNA methylation profiles of atherosclerosis patients enrolled in the Left Ventricular Assist Device Study (LVAD), Istas and colleagues (2017) conducted genome-wide DNA methylation sequencing using the Illumina Infinium HumanMethylation450 array, in the whole blood of eight atherosclerosis patients and eight healthy volunteers. After adjusting for age, they identified 287 hypo- and 229 hypermethylated CpG sites in cases versus controls, which mapped to genes involved in the cell cycle, cell adhesion, and cell death [26]. They validated the DNA methylation of nine CpG sites in *BRCA1 DNA repair associated* 1 (*BRCA1*) and seven CpG sites in *cysteine-rich secretory protein 2* (*CRISP2*), using an Epityper MassARRAY approach. In addition, they identified associations between three CpG sites in *CRISP2* (cg12440062, cg25390787, and cg01076129) and one CpG site in *BRCA1* (cg16630982), as well as subclinical atherosclerosis measures in an independent cohort comprising 24 participants from the Aragon Workers Health Study (AWHS). While *BRCA1* is widely recognized as a tumor suppressor, recent evidence suggests that it may act as an anti-inflammatory protein to retard endothelial dysfunction and atherosclerosis [122]. Data on *CRISP2* methylation and its role in atherosclerosis is a novel concept.

Using the Illumina Infinium HumanMethylation850K BeadChip array, Antoun (2022) investigated DNA methylation signatures in the whole blood of 293 Gambian and 698 Indian children. Several differentially methylated regions were identified for each cohort and were associated with systolic blood pressure [annotating to *mediator complex subunit 22* (*MED22*, g13455829) and *Egl-9 family hypoxia inducible factor 2* (*EGLN2*, cg22671726)], fasting insulin [annotating to the gene bodies of *family with sequence similarity 46 member A* (*FAM46A*, cg22388948) and *LysM domain containing 2* (*LYSMD2*, cg13934266)], circulating triglycerides (cg15237100 located in an intergenic region), and LDL-cholesterol (cg01469688 and cg06952751, mapping to the promoter regions of *suppressor of cancer cell invasion* (*SCAI*) and *C18orf8*, respectively) in the Gambian cohort, and with insulin sensitivity [mapping to the gene body of *transmembrane protein 57* (*TMEM57*, cg10304969)] and HDL-cholesterol [cg04988216 annotating to the gene body of *Receptor Tyrosine Kinase Like Orphan Receptor 1* (*ROR1*)] in the Indian cohort. Interestingly, the differentially methylated CpGs identified in the study were unique to each cohort and no overlap was observed for any CpGs between groups. Since DNA methylation at specific CpGs can be influenced by genotype, the authors also identified single nucleotide polymorphisms (SNPs) within an approximately 2 MB region from the CpGs of interest that were associated with the differential methylation. Significant quantitative trait loci (methQTLs) were found for three differentially methylated CpGs associated with LDL-cholesterol in Gambians; however, methylation did not mediate genotype effects on CVD outcomes in the Indian population [55]. Therefore, the authors argued that DNA methylation patterns could be population-specific, and that environmental and ethnic differences could influence these changes.

Agha (2019) conducted a longitudinal multi-cohort epigenome-wide search for DNA methylation alterations associated with CHD events across nine different cohorts, comprising 11273 study participants from Europe and the United States. A total of 442192 CpGs were analyzed for all nine cohorts, of which DNA methylation at 52 CpG sites was associated with CHD and myocardial infarction incidence. These CpG sites mapped to key genes associated with calcium flux [including those mapping to *ATPase Plasma Membrane Ca2+ Transporting 2* (*ATP2B2*), *calcium sensing receptor* (*CASR*), *guanylate cyclase activator 1B* (*GUCAIB*), and *hippocalcin like 1* (*HPCAL1*)], arterial plaque calcification [such as *protein tyrosine phosphatase receptor type N2* (*PTPRN2*)], and kidney function [including CpG sites mapping to *cadherin related 23* (*CDH23*) and *hippocalcin-like 1* (*HPCAL1*)]. Interestingly, and in contrast to the findings of Antoun (2022), the associations between the 52 CpG sites and CAD incidence were consistent across all cohorts in the study. Moreover, these associations were not influenced by ethnicity, as they remained similar when comparing European groups to those of African American ancestry [56].

Using reduced representation bisulfite sequencing (RRBS), Li et al. (2017) assessed the DNA methylation profile in the blood leukocytes of 27 heart failure patients compared to 20 healthy controls. They identified three genes, *solute carrier family 2 member 1* (*SLC22A1*), *MPV17 mitochondrial inner membrane protein like* (*MPV17L*), and *plectin* (*PLEC*), as differentially methylated in the heart failure patients compared with the healthy controls, with the DNA methylation of these genes being inversely associated with their mRNA expression levels [57]. While the role of *SLC22A1* in heart failure is not well established, it has been shown to promote tumor cell proliferation and glycolysis in cancer [123]. An increase in glycolysis is believed to reduce cardiac hypertrophy and dysfunction induced by pressure overload, suggesting that *SLC22A1* may be beneficial in cardiac function, but this requires further investigation [124]. On the other hand, *MPV17L* has previously been shown to prevent pancreatic β-cell hyperplasia, suggesting that the protein may play a critical role in preventing diabetes-related cardiac dysfunction [125]. The *PLEC* gene encodes a cytoskeletal linker protein that has been previously implicated in cardiomyopathy [126].

Using the Illumina HumanMethylation450K BeadChip system, Huan (2019) studied the methylome of 4170 participants of European ancestry enrolled in the FHS, after which findings were replicated in two independent populations from the Atherosclerosis Risk in Communities (ARICs) and Grady Trauma Project (GTP) study cohorts. Their collective findings highlighted DNA methylation at 92 CpGs as a putative causal factor for cardiovascular disease and myocardial infarction, with some of the CpGs annotating to genes linked to heart failure [58]. Indeed, *lysosomal acid lipase* (*LIPA*) has been linked to hypercholesterolemia [127], whilst *ABO* and *serologically defined colon cancer antigen 8* (*SDCCAG8*) have been identified as genes underlying the development of CHD and diastolic blood pressure, respectively [128,129].

Using monozygotic and dizygotic twin pairs from a Swedish population, Qin et al. (2021) identified 20 top-ranked CpG sites associated with non-stroke CVDs, overall stroke, and ischemic stroke [59]. The methylation of these CpGs was shown to determine the levels of cardiometabolic traits, such as BMI and blood pressure, with the latter mediating CVD risk. Similarly, using RRBS, Koseler and colleagues assessed the DNA methylation differences of thousands of CpG sites between one pair of monozygotic twins discordant for CVDs [60]. Here, DNA methylations, annotating to the genes involved in fatty acid and cholesterol metabolism [namely, *lipid droplet associated hydrolase* (*LDAH*), *apolipoprotein B* (*APOB*), *acyl-CoA synthetase* members 2A, 5, and 3 (*ACSM2A*, *ACSM5*, and *ACSM3)*, *carboxylesterase 1* (*CES1*), *carboxylesterase 1 pseudogene 1* (*CES1P1*), *AFG3 like matrix AAA peptidase subunit 2* (*AFG3L2*), *iron-sulfur cluster assembly enzyme* (*ISCU*), *SEC14 like lipid binding 2* (*SEC14L2*), and *microsomal triglyceride transfer protein* (*MTTP*)], were hypomethylated in the healthy twin as compared with the twin presenting with premature CAD. Some of these genes, including *APOB*, have been previously reported to be strong drivers of myocardial infarction risk [130]. The authors noted that the DNA methylation differences observed between the twins could be attributed to the influence of lifestyle or statin treatment [60].

To identify the association between DNA methylation and myocardial infarction, Nakatochi and colleagues (2017) used samples collected from the whole blood of elderly male Japanese subjects with a history of myocardial infarction, as compared with a control group. The control group was chosen from an ongoing Kita-Nagoya Genomic Epidemiology (KING) study and was selected based on their nondiabetic status [61]. Data revealed an association between two CpG sites [mapping to *zinc finger homeobox 3* (*ZFHX3*, cg07786668) and *SWI/SNF-related matrix-associated actin-dependent regulator of chromatin subfamily a member 4* (*SMARCA4*, cg17218495)] and myocardial infarction, which was independent of CVD risk factors, such as BMI, blood lipid levels, and type 2 diabetes. While *ZFHX3* has been implicated in the pathophysiology of obesity, *SMARCA4* has been shown to regulate cholesterol levels, suggesting that the methylation of both genes could be essential biomarkers of myocardial infarction risk [131,132].

Rask-Andersen et al. (2016) conducted an epigenome-wide association study to identify differentially methylated genes in the peripheral blood from participants from the Northern Sweden Population Healthy Study cohort, comprising individuals with a history of myocardial infarction. They identified altered methylation at 211 CpG sites, which correlated with myocardial infarction, a correlation that remained independent of BMI [44]. Some of these CpG sites annotated to genes previously implicated in cardiovascular diseases, including *axonally derived neuregulin 1* (*NRG1*), known to play a role in myocardial repair following myocardial infarction [44]. Moreover, they reported that the methylation and expression of *dysferlin* (*DYSF*), encoding a protein involved in membrane integrity of skeletal muscle cells, provided a strong predictive value for atherosclerotic cardiovascular disease in their cohort [133,134]. Finally, Ward-Caviness and colleagues (2018) attempted to develop an epigenetic fingerprint predictive of myocardial infarction incidence in the Cooperative Health Research in the Augsburg Region (KORA) cohort [62]. Using the Illumina Infinium HumanMethylation450 BeadChip, they identified the methylation of 11 CpGs that related to incidents of myocardial infarction, although following postadjustment for medication, only 9 CpGs retained statistical significance. Importantly, these modifications were shown to be generalized across various ethnicities and geographical regions. Of these, two CpGs were annotated to low-density *lipoprotein receptor-related protein 8* (*LRP8*) and *potassium calcium-activated channel subfamily N member 1* (*KCNN1*), genes shown by others to be associated with CVDs [135,136]. Indeed, a previous study showed that phosphorylation-induced activation of *LRP8*, upon binding to *LDL*, leads to the activation of p38 mitogen-activated protein kinase, a critical protein involved in myocardial injury during ischemia and reperfusion [135]. The *KCCN* genes encode for small-conductance Ca2+-activated K+ channels that are known to play a key role in cardiac excitability [137].

## 3. Future Perspectives and Conclusions

The pathogenesis of CVDs involves a complex interaction between genetic, environmental, and lifestyle factors, and there has been some success in identifying DNA methylation alterations that are associated with these diseases. Screening individuals using non-invasive methods may represent opportunities for disease prevention and intervention. The current review aimed to provide a comprehensive analysis of the latest reported DNA methylation alterations as blood-based biomarkers for CVDs. The review included forty-eight studies covering various CVD phenotypes, conducted in five continents (Europe, Asia, North and South America, and Africa), featuring diverse populations, age groups, blood cell types, and study designs. While no strong consensus exists on the association between global DNA methylation and CVDs, candidate gene and genome-wide studies collectively identified a repertoire of over 100 genes with methylation statuses associated with CVDs, their risk factors, or related conditions (Table 4). These genes included *FOXP3*, *IL-6*, *CPT1A*, *ABCG1*, and *AHRR*, which were differentially methylated in individuals with CVDs compared to healthy individuals in at least two or more studies. Moreover, these five genes were consistently differentially methylated across various populations, methodological approaches, and blood cell types, supporting their candidacy as biomarkers of CVDs.

Most studies covered in this review performed their analyses in whole blood, utilized repetitive elements as proxies for quantification of global DNA methylation, applied bisulfite sequencing or pyrosequencing for candidate gene studies, and employed the Illumina Infinium HumanMethylation450K for measuring DNA methylation in genome-wide studies. Due to practical and ethical reasons, the procurement of tissues central to CVD development, such as biopsies of the myocardium or blood vessel walls, remains difficult due to the invasive procedures required for their collection. In this regard, peripheral blood can be collected using minimally invasive, inexpensive methods and has been shown to be a reliable source of DNA methylation information, possibly due to the cardiovascular system being significantly regulated via the blood and its constituents. Indeed, a previous study reported a high correlation between the DNA methylation in the peripheral blood leukocytes and atrial biopsies collected from patients undergoing coronary bypass surgery [138], thus supporting the utility of blood-based DNA methylation as a proxy for cardiac cells. However, it is important to note that whole blood contains various cell types, which may lead to variability in DNA methylation signatures, based on inter-individual differences in the fraction of DNA-containing blood cells collected in studies. There are several methods that can be used to account for these confounding effects, including the flow-cytometry-based sorting of blood cells and subsequent quantification of the DNA methylation in the resulting individual cell types, or the post hoc adjustment of cell count based on published regression models, such as that reported by Houseman et al. (2012) [139].

The global DNA methylation studies covered in this review presented contradictory findings, with DNA hypomethylation associated with CVD events in six studies, hypermethylation associated with the same CVD traits in five studies, and one study reporting no significant differences in the global DNA methylation levels in individuals with atherosclerosis compared to healthy controls. These reports are consistent with other reviews on the relationship between DNA methylation and metabolic diseases, including dyslipidaemia, type 2 diabetes, and CVDs [64,140,141,142]. These inconsistencies might be explained, in part, by heterogeneous methodologies applied by the different studies, including differences in sample size, methods used for DNA isolation and methylation quantification, and biological sources (i.e., whole blood versus PBMCs), as well as adjustments for confounding factors. Moreover, it is widely accepted that the influence of DNA methylation on gene expression is dependent on its genomic location, and thus, methylation at specific disease-causing genes/loci may be a more informative measure of health-related outcomes compared to the global, pooled analysis of methylated cytosines within the genome.

Most of the studies included in this review utilized either a candidate gene or a genome-wide approach. For candidate gene studies, the DNA methylation of targeted regions was selected based on hypotheses generated from previous genetic or biological studies. Only two genes, *IL-6* and *FOXP3*, were examined and replicated in more than one study; therefore, most of the remaining candidate gene findings require additional replication. *IL-6* encodes a pro-inflammatory cytokine and a growing list of studies have implicated this gene in CVD events [143]. It is speculated that, under atherosclerotic conditions, hypomethylation of the *IL-6* promoter would cause upregulation of its expression, leading to a higher production of the cytokine by macrophages and consequent vascular inflammation. *FOXP3* encodes a transcription factor that is responsible for lineage specification, cell maturation, and the function of regulatory T-cells (Tregs). Studies in numerous experimental models have shown that Tregs play a powerful protective role in atherosclerosis. For example, Kligenberg and colleagues (2013) showed that the depletion of FOXP3+ Treg cells resulted in a 2.1-fold increase in plaque size and hypercholesterolemia in mice [144]. We, therefore, postulate that in humans, hypermethylation of *FOXP3*, as observed by Zhu (2019) and Jia (2013), and reported in this review, may result in suppression of the gene and concomitant depletion of Treg cells, resulting in a decreased protective effect of these cells against the development of atherosclerosis [31,32].

This review also included 22 genome-wide DNA methylation studies. This approach has become increasingly popular and preferred over candidate gene methods as it is more effective in detecting novel disease-associated DNA methylation marks. However, the drawbacks include increased cost, the need for large sample sizes, as well as the requirement for validation in independent replication cohorts [145]. Most genome-wide studies covered here validated their results in one or more replication cohorts, however, some used relatively small sample sizes. Amongst the differentially methylated loci identified, CpGs mapping to three genes (cg00574958 in intron 1 of *CPT1A*, cg27243685 in the 5′UTR of *ABCG1*, and cgo5575921 in the gene body of *AHRR*) were differentially methylated in more than one genome-wide study. Moreover, while the differentially methylated CpG sites within *CPT1A* and *ABCG1* do not reside in promoter regions, we note that these patterns were reported to be accompanied by the altered expression of these genes, suggesting that gene expression in this context, may be regulated by methylation at distal sites.

*CPT1A*, encoding a constituent of the carnitine palmitoyltransferase (CPT) system, is an important rate-limiting enzyme involved in the conversion of cytoplasmic long-chain acetyl-CoA to acylcarnitine, which in turn, enters the mitochondria for fatty acid β-oxidation (FAO) [146]. FAO is crucial for endothelial cell metabolism as it aids in resisting oxidative stress in the development of blood vessels. It is therefore not surprising that CPT1A has been proposed as a candidate target for the treatment of vascular-related diseases. In mice, the inhibition of CPT1A activity, and the consequent reduction of FAO, resulted in improved whole-body glucose tolerance and insulin sensitivity [147]. In addition, a CPT1A inhibitor, etomoxir, showed efficacy in improving outcomes for heart failure patients in a clinical trial; however, the trial was terminated early due to unacceptably high liver transaminase levels in enrolled patients [148]. *ABCG1* encodes a protein involved in the efflux of cholesterol to high-density lipoprotein (HDL) in macrophages. As observed by Hedman (2017) and Mendelson (2017), hypermethylation of one CpG site located in the 5′UTR of *ABCG1*, as well as the concomitant reduced expression of *ABCG1* mRNA, was associated with cardiovascular risk in the FHS, PIVUSs, GOLDN, LBCs, and ARICs cohorts [51]. We speculate that the downregulation of *ABCG1* may contribute to CVD risk by lowering the capacity for reverse cholesterol transport, potentially leading to the accumulation of excessive cholesterol in the body and the development of atherosclerotic plaques [149]. The *AHRR* gene encodes for a repressor of the aryl hydrocarbon receptor, which amongst other roles, inhibits the metabolism of polycyclic aromatic hydrocarbons, a carcinogen found in cigarette smoke [150]. Interestingly, the DNA methylation of cg05575921 within *AHRR* is a well-documented marker of cigarette smoking, as well as overall mortality risk [151]. Two studies covered in this review implicated the DNA methylation of cg05575921 with CVD risk, both of which adjusted their data for smoking status. We speculate that the connection between cg05575921 and CVD risk in these studies may be a true association, or it may be due to residual confounding with smoking, failures in self-reporting of smoking, and/or inter-individual sensitivities to smoking, as well as its lasting biological effects.

Some limitations of this review should be acknowledged. First, whilst effort was made to conduct a comprehensive literature search, we cannot disregard the possibility of publication bias due to underreporting of negative findings. Our literature search was also conducted in only three databases and excluded studies that were not published in English. Most reviewed studies utilized a cross-sectional design, and due to the unstable nature of the epigenome, we cannot ascertain whether the identified signatures are a cause or consequence of CVDs. Moreover, only DNA methylation modifications in CVDs were reviewed and we therefore cannot rule out the effect of other epigenetic marks, such as posttranslational histone modifications and non-coding RNAs, as biomarker candidates of CVDs. A key limitation of many of the reviewed studies was the use of small sample sizes and a lack of adjustment for potential confounders. It is well established that factors such as sex, age, smoking, alcohol consumption, comorbidities, and medication use significantly influence DNA methylation profiles across numerous genes [152,153,154]. However, among the 46 studies analyzed, only 33 accounted for age. Ten studies were conducted exclusively in males, three exclusively in females, and three did not specify the participants’ sex. Of the 32 studies that included both sexes, only 23 adjusted for sex. The studies also encompassed a range of CVD phenotypes and methodological approaches, posing challenges for comparative analysis. Nevertheless, the DNA methylation of FOXP3, IL-6, CPT1A, ABCG1, and AHRR was consistently associated with CVDs across multiple studies. Importantly, each of these studies accounted for age, sex, and additional covariates, such as BMI, smoking, comorbidities, and blood cell count, reinforcing the reliability of these findings and supporting their potential as CVD biomarkers.

Our review also has several strengths. Due to the difficulty in accessing tissues central to the pathogenesis of CVDs, our review focused on associations between blood-based DNA methylation signatures and CVDs, and managed to consolidate findings from 48 research articles, identifying *FOXP3*, *IL-6*, *CPT1A*, *ABCG1*, and *AHRR* as promising biomarker candidates for CVDs (Figure 2). These findings may have important implications for the development of non-invasive epigenetic risk assessment tests for screening individuals for CVDs. We also highlighted differentially methylated genes that were associated with major risk factors for CVDs, including HDL/LDL-cholesterol levels, triglyceride concentrations, and BMI, as well as HbA1c and glucose levels, further supporting the notion that blood-based screening for DNA methylation could be utilized to monitor individuals who are at high risk of developing CVDs and who require medical and lifestyle interventions. In this context, the DNA methylation candidates identified in this review should be further evaluated in large-scale longitudinal studies using high-quality DNA methylation quantification platforms, with careful consideration of confounders, such as age and sex. To date, DNA methylation biomarkers for CVDs have yet to be assessed in a clinical trial setting, however, we anticipate that the continuous advancement of next generation sequencing methods will expedite the identification of novel predictive CVD biomarkers, and when implemented, will have a sizable public health impact.

## Figures and Tables

**Figure 1 ijms-26-02355-f001:**
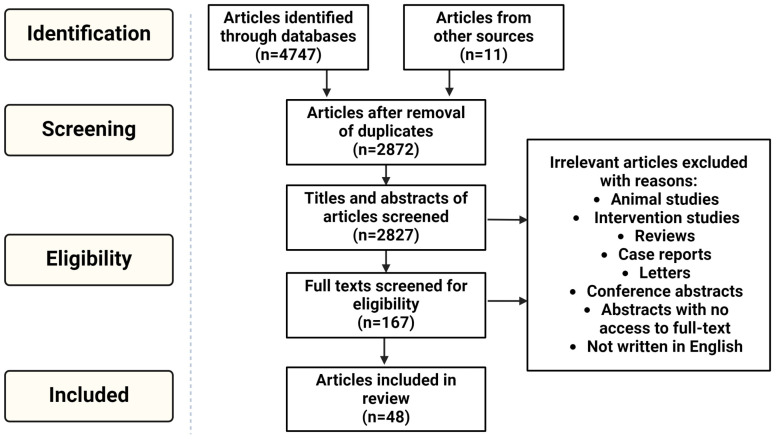
Flow diagram depicting the selection of research articles for inclusion in the review. Adapted from [18].

**Figure 2 ijms-26-02355-f002:**
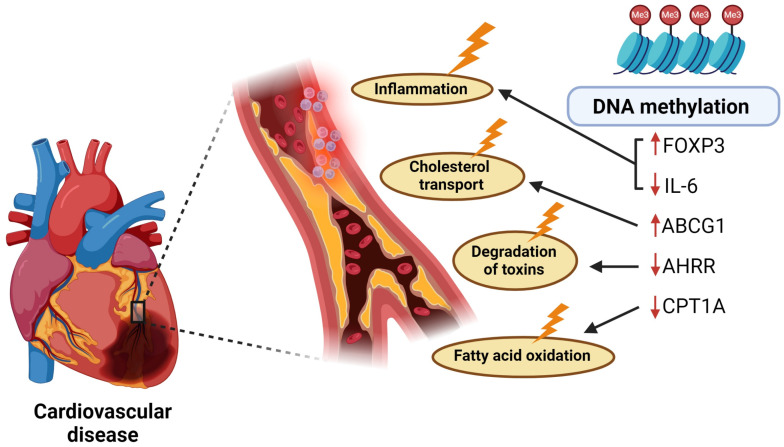
DNA methylation alterations associated with cardiovascular disease.

**Table 1 ijms-26-02355-t001:** Main findings from CVD studies using global DNA methylation analysis in peripheral blood.

Author	Population	Tissue Type	Method	Study Outcome
[19]	European; 7 male atherosclerotic vascular patients, 7 healthy male controls	WBCs	5mC	Atherosclerosis patients had significantly higher plasma homocysteine concentrations and significantly lower genomic DNA methylation.
[20]	Indian; 137 coronary artery disease patients, 150 controls	PBLs	HpaII/MspI ratio	Global hypermethylation was associated with coronary artery disease.
[21]	American; 712 elderly men	WB	LINE-1	Blood LINE-1 hypomethylation was associated with ischemic heart disease.
[10]	Singapore Chinese; 129 men, 157 women	PBLs	ALU and Satellite 2 repetitive element methylation	Positive correlation in global DNA methylation in men with a history of CVD or its predisposing conditions but not in women.
[22]	Chinese; 280 ischemic stroke patients and 280 healthy controls; 40 and 80 years	WB	LINE-1	Global hypomethylation was associated with higher risk for ischemic stroke in men but not in women.
[23]	Chinese; 334 CHD patients and 788 healthy controls	PBLs	LINE-1	LINE-1 hypomethylation was significantly associated with age, elevated total cholesterol, diagnosis of diabetes, and risk of CHD.
[24]	Italian European; 292 myocardial infarction patients and 292 matched controls	WBCs	LINE-1	Global hypomethylation was associated with myocardial infarction risk in men.
[25]	Brazilian; 90 postmenopausal women between 45 and 65 years	PBLs	5mC	Hypomethylation was associated with higher CVD risk.
[26]	German; 8 atherosclerosis patients and 8 healthy controls; 35–60 years	WB	5mC/5hmC	No statistically significant changes in global DNA methylation in atherosclerosis patients compared to healthy individuals.
[27]	American; longitudinal study of 789 elderly subjects followed over 10 years	WB	LINE-1 and Alu	Positive association identified between Alu hypermethylation and diastolic blood pressure.
[28]	Chinese; 91 atherosclerosis patients and 22 controls	PBMCs	5mC/5hmC	5mC and 5hmC levels were significantly correlated with Gensini (severity of atherosclerosis).
[29]	Chinese; 44 patients with cardiovascular disease and 42 controls	PBMCs	5mC/5hmC	5mC and 5hmC levels were higher in CVD patients than in control subjects.

**Table 2 ijms-26-02355-t002:** Main findings from CVD studies investigating candidate gene methylation in peripheral blood.

Author (Year)	Genes Investigated	Country	Sample Size	Sex	Tissue Type	Method	Study Outcome
[27]	*TLR2*, *IFN-γ*, and *iNOS*	United States	Longitudinal study of 789 subjects	M	WB	Bisulphite pyrosequencing	Methylation of *TLR2* and *iNOS* positively correlated with blood pressure, whilst *IFN-γ* promoter methylation negatively associated with blood pressure.
[30]	*BMPR2*	China	VHD = 82 Controls = 57	M and F	PB	Bisulphite pyrosequencing	The *BMPR2* promoter was hypermethylated in the VHD patients compared to healthy controls and correlated with VHD severity.
[31]	*FOXP3*	China	171 patients with CHD	M	PB	Methylation specific PCR	Increased DNA methylation of *FOXP3* associated with severity of atherosclerosis.
[32]	*FOXP3*	China	CHD = 188 Controls = 68	M	WB	Bisulfite sequencing	Increased DNA methylation of *FOXP3* associated with severity of CHD.
[33]	*microRNA-510*	India	Hypertension = 20 Controls = 20	M and F	WB	Methylation specific PCR	The *miR-510* promoter was hypermethylated in the hypertensive patients versus healthy controls.
[34]	*FABP3* (17 CpGs)	Europe	517 individuals with varying parameters of metabolic syndrome	M and F	PBMCs	EpiTYPER MassArray	Average methylation of 17 CpG sites within the *FABP3* positively associated with total cholesterol, whilst individual CpG methylation associated with diastolic blood pressure, insulin resistance, and LDL-cholesterol.
[35]	*hTERT* (5 CpGs)	China	Atherosclerosis = 197 Controls = 165	M and F	PBLs	Methylation-specific PCR	*hTERT* was hypomethylated in atherosclerosis patients compared to healthy individuals and associated with a concomitant decrease in *hTERT* mRNA levels.
[36]	*IL-6* (2 CpGs)	China	CHD = 212 Controls = 218	M and F	PBLs	Bisulfite pyrosequencing	Hypomethylation at two promoter CpGs in CHD patients compared to controls. Combined methylation scores of both CpGs correlated with CHD risk.
[37]	*miRNA-233* (9 CpGs)	China	Atherosclerosis = 23 Controls = 32	M and F	PBLs	Bisulfite sequencing	Hypomethylation of seven CpG sites were detected in the *miRNA-233* promoter in participants with atherosclerotic cerebral infarction compared to healthy controls. DNA methylation of these CpG sites correlated with total plasma cholesterol levels.
[38]	*NR3C1* (22 CpGs)	United States	84 MZ twin pairs	M	PBLs	Bisulfite pyrosequencing	Altered DNA methylation in exon 1F of the *NR3C1* promoter positively associated with markers of early subclinical atherosclerosis.
[39]	*IL-6* (6 CpGs)	Iran	Atherosclerosis = 35 Controls = 30	M and F	PB	Bisulfite sequencing	Six evolutionary conserved CpG sites in the distal *IL-6* promoter were hypomethylated in the atherosclerosis patients compared to the controls. DNA methylation of these CpG sites inversely correlated with *IL-6* mRNA levels.
[40]	*NPPA* (9 CpGs)	China	Longitudinal study of 2498 subjects	M and F	PB	Bisulfite sequencing	Hypermethylation of a single CpG site situated −459 bp from the *NPPA* TSS associated with reduced risk of developing CVDs.
[41]	*SOAT1* (26 CpGs)	China	CHD = 99 Controls = 89	M and F	WB	Bisulfite sequencing	Hypomethylation of seven CpG sites associated with incidence of CHD.
[42]	*TXNIP* (1 CpG)	China	CAD = 54 Controls = 54	M and F	PBLs	Pyrosequencing	DNA methylation of *TXNIP* at a single CpG site, located at cg19693031, was significantly decreased in cases compared to controls. DNA methylation inversely correlated with *TXNIP* mRNA expression, fasting plasma glucose, and total cholesterol.

**Table 3 ijms-26-02355-t003:** Main findings from CVD studies investigating whole-genome methylation in blood.

Author (Year)	Country	Sample Size	Sex	Tissue Type	Method	Study Outcome
[43]	India	Discovery cohort CHD = 18 Controls = 18 Validation cohort CHD = 48 Controls = 48	M	PBMCs	Bisulphite sequencing via 454 platform	Hypermethylation was identified within 19 regions in CHD patients versus controls. Twelve differentially methylated regions were assessed in the validation cohort, which identified the hypermethylation of 6 CpG sites in the CHD group compared to controls. Three of these sites mapped to the intronic region of *STE20 related adaptor alpha* (*STRADA*), one was situated in the intronic region of *heat shock protein 90 beta family member 3 pseudogene* (*HSP90B3P*), and two were situated within exon 1 of *Complement C1q like 4* (*C1QL4*).
[24]	Italian and Dutch	Discovery cohort MI = 292 Controls = 292 Validation cohort MI = 317 Controls = 262	M and F	WBCs	Discovery: Illumina Infinium HumanMethylation450 BeadChip Validation: MALDI-TOF mass spectrometry	Results from the discovery cohort revealed the hypomethylation of 15 CpG sites within exon 1 of the *zinc finger and BTB domain-containing protein 12* (*ZBTB12*) gene, which correlated with MI, and was replicated in the validation cohort.
[44]	Swedish	CVD = 238 Controls = 491	M	PB	Illumina Infinium HumanMethylation450 Beadchip	Altered DNA methylation was observed at 211 CpG sites, mapping to 196 genes, which collectively associated with a history of MI. Forty-two of the differentially methylated regions mapped to genes previously associated with MI in the literature. Some of these genes identified included *Dysferlin* (*DYSF*) and *Neuregulin 1* (*NRG1*). The link between MI and methylation was independent of BMI.
[45]	European and African	Stroke patients = 180	M and F	WB	Illumina Infinium HumanMethylation450 BeadChip	Hypermethylation at two CpG sites located in *Ankyrin Repeat And SOCS Box Containing 10* (cg22812874, *ASB10*) and *Tetratricopeptide repeat domain 37* (cg00340919, *TTC37*) associated with plasma homocysteine levels in individuals with recurrent stroke. This association was found in Europeans but not in Africans.
[46]	American and European	Discovery REGICOR cohort n = 645 Validation FOS cohort n = 2542	M and F	WB	Illumina Infinium HumanMethylation450 BeadChip	Eight CpG sites annotating to four genes [*Aryl hydrocarbon receptor repressor* (*AHRR*), *Carnitine Palmitoyltransferase 1A* (*CPT1A*), *Peptidylprolyl Isomerase F* (*PPIF*), and *Strawberry Notch Homolog 2* (*SBNO2*)], and three intergenic regions were differentially methylated and associated with cardiovascular risk, independent of age.
[47]	European	T2D + HF = 10 Control = 10	M	WB	Discovery: Methyl-binding domain-capture sequencing Validation: Bisulfite sequencing	Differential methylation was identified in both gene body and enhancer elements, with reduced methylation signatures observed in coding regions of [*Histone deacetylase 9* (*HDAC9*), *MicroRNA 3675* (*MIR3675*), *Jumonji- and AT-rich interaction domain (ARID)-domain-containing protein* (*JARID2*), and *Gremlin 1* (*GREM1*)], and with increased methylation in *Decaprenyl Diphosphate Synthase Subunit 2* (*PDSS2*), linked to the extreme phenotypes of heart failure.
[48]	Chinese	CVD = 83 Controls = 83	M and F	WB	Illumina Infinium HumanMethylation850 EPIC BeadChip	Eight CVD-related CpGs sites were identified, with the combined use of all eight showing an excellent predictive power for CVD development. The 8 CpGs included: 1 hypermethylated CpG of *cytochrome P450*, *family 8*, *subfamily B*, *and polypeptide 1* (cg07655795, *CYP8B1*), being associated with increased CVD risk, whilst the remaining seven hypomethylated CpGs of *Spondin 1* (cg06901278, cg11651314, and *SPON1*), *Phosphofurin acidic cluster sorting protein 1* (cg03914662, *PACS1*), *Uveal Autoantigen With Coiled-Coil Domains And Ankyrin Repeats* (cg09306458, *UACA*), *Coiled-coil domain containing 50* (cg05359217, cg05946546, *CCDC50*), and *Haemagglutinin stalk domain 17811* (cg02518222, *HASD17811*) were associated with an increased CVD risk.
[49]	African American and European	Atherosclerosis = 93	M and F	WB	Illumina Infinium HumanMethylation850 EPIC BeadChip	Four CpGs were associated with atherosclerosis after adjusting for CVD risk. The CpGs included *Aryl hydrocarbon receptor repressor* (cg05575921, cg21161138, *AHRR*), *Growth differentiation factor 11* (cg09936388, GF11), and *Leucine Rich Repeat Containing 52* (cg181168448, *LRRC52*).
[50]	American and European	Discovery cohort (REGICOR-1) MI = 191 Controls = 195 Validation cohort (REGICOR-2) MI = 101 Controls = 103 Follow-up association studies WHI = 1863 FOS = 2540	M and F	WB	Illumina Infinium HumanMethylation850 EPIC BeadChip	A total of 34 of the identified CpG sites associated with MI in both REGICOR-1 and REGICOR-2 samples. Of these, 25 were located within gene-coding regions, whereas 9 mapped to intergenic regions. Importantly, 3 of the identified CpG sites associated with CVD in follow up association studies using the WHI and FOS cohorts. The 3 identified CpGs were *Aryl hydrocarbon receptor repressor* (cg05575921, *AHRR*), *F2R like thrombin or trypsin receptor 3* (cg21566642, *F2RL3*), and *myeloperoxidase* (cg04988978, *MPO*). More importantly, DNA methylation changes in cg21566642 showed a genetic influence in childhood (rs139595493) and adolescence (rs72617176).
[51]	American and European	Discovery cohorts FHS =1494 PIVUSs = 812 Validation cohorts LBCs1921 ≤ 380 LBCs1936 ≤ 654 GOLDN = 991	M and F	WB	Illumina Infinium HumanMethylation450 BeadChip	Differential methylation was identified at 33 CpG sites in 5 different study cohorts. Of these, 25 CpG sites showed novel associations with circulating lipid traits (HDL, LDL, and TGs), including a CpG site annotated to the *ATP Binding Cassette Subfamily G Member 1* locus (cg27243685, *ABCG1*). Methylation of this site correlated with a 38% increased risk of CHD.
[52]	USA, UK, Africa, Europe, Sweden	Discovery cohorts FHS = 2377 LBCs = 1366 Validation cohorts ARICs = 2096 GOLDN = 992 PIVUSs = 967	M and F	WB and CD4+ cells	Illumina Infinium HumanMethylation450 BeadChip	Differential methylation in 135 CpGs were identified in the discovery cohorts, of which 83 could be replicated in the three validation cohorts. The study reported a three-way association between 11 differentially methylated CpGs, expression of 7 genes, and BMI. These associations included the DNA methylation and expression status of *ATP Binding Cassette Subfamily G Member 1* (cg01881899, *ABCG1*), Carnitine palmitoyltransferase 1A (cg00574958, cg17058475, *CPT1A*), and *Sterol regulatory element-binding protein 1* (cg11024682, *SREBF1*).
[53]	Canadian	Discovery cohort IUCPQ = 206 Validation cohort CaG = 137	M and F	VAT WB	Methyl-binding domain-capture sequencing	This study identified 565 differentially methylated regulatory regions in adipose tissue, which were associated with plasma lipid levels. Of these, 340 were reflected in the whole blood, of which 68 (mapping to 13 putative enhancers and 55 promoters) shared similar associations with plasma lipid traits, including *serine/threonine kinase 1* (AKT1), *histone deacetylase 4* (*HDAC4*), *Bone morphogenetic protein 4* (*BMP4*), *Growth differentiation factor 7* (*GDF7*), *Ceramide Kinase* (*CERK*), *Vestigial-like family member 3* (*VGLL3*), and *ATP binding cassette subfamily C member 5* (*ABCC5*). Of the 68 CpG sites identified in the IUCPQ cohort, 22 could be validated in the CARTaGENE cohort.
[54]	Ghanaian	T2D = 589	M and F	WB	Illumina Infinium HumanMethylation450 BeadChip	The study identified 14 novel differentially methylated loci associated with CRP levels up to 40 mg/L. Three of the differentially methylated CpGs, annotating to *Pyruvate Carboxylase* (cg14653250, *PC*), BTG Anti-Proliferation Factor 4 (cg13767940, *BTG4*), and *Peptidyl Arginine Deiminase 1* (cg21010178, *PADI1*), showed an association with CVD risk.
[26]	German	Discovery cohort: Atherosclerosis = 8 Controls = 8 Validation cohort: AWHS = 24	M and F	WB	Discovery: Illumina Infinium HumanMethylation450 BeadChip Validation: Epityper MassARRAY	Differential methylation was identified in CpGs (287 hypo- and 229 hypermethylated) in cases versus controls which related to genes involved in cell cycle, adhesion, and death, among others. DNA methylation of *breast cancer 1* (*BRCA1*) and *Cysteine Rich Secretory Protein 2* (*CRISP2*) were validated using an Epityper MassARRAY and showed, in the additional AWHS cohort, that three CpG sites in *CRISP2* (cg12440062, cg25390787, cg01076129) and one CpG site in *BRCA1* (cg16630982) associated with subclinical atherosclerosis measures.
[55]	Gambian and Indian	n = 293 Gambian n = 698 Indian	M and F	WB	Illumina Infinium HumanMethylation850 BeadChip	Several differentially methylated regions were associated with diastolic blood pressure, insulin sensitivity, triglycerides, and LDL-cholesterol in the Gambian population, and insulin sensitivity and HDL-cholesterol in the Indian population. Importantly, there was no overlap of differentially methylated CpGs identified between the cohorts. The study identified significant quantitative trait loci (cis-methQTLs) at three LDL-cholesterol-associated differentially methylated CpGs in Gambians; however, methylation did not mediate genotype effects on the CVD outcomes in the Indian population.
[56]	American and European	ARICs = 2567 CHS = 197 EPICORs = 584 FHS = 2375 InCHIANTI = 457 KORA = 1377 NAS = 484 WHI-EMPC = 1610 WHI-BAA23 = 1622	M and F	PBLs	Illumina Infinium HumanMethylation450 BeadChip	An association was found between methylation at 52 CpG sites and the risk of developing CHD or myocardial infarction in nine population-based cohorts. These CpG sites mapped to key regulatory regions of genes associated with calcium flux [*ATPase Plasma Membrane Ca2+ Transporting 2* (*ATP2B2*), *calcium sensing receptor (CASR)*, *guanylate cyclase activator 1B* (*GUCAIB*), and *hippocalcin like 1* (*HPCAL1*)], arterial plaque calcification [Protein Tyrosine Phosphatase Receptor Type N2 (PTPRN2)], and kidney function [Cadherin Related 23 (CDH23) and Hippocalcin-like 1 (HPCAL1)]. Identified loci remained significant across all nine cohorts and significance was not influenced by race.
[57]	China	HF = 27 Controls = 20	M and F	PBLs	Reduced Representation Bisulfite Sequencing	Assessment of DNA methylation in African American males younger or equal to 30 years of age revealed an increase in methylation of two CpGs in the *Sulfatase 1* (*SULF1*) gene in HF patients, as compared to age-matched normotensive controls.
[58]	European and African American	Discovery cohorts FHS offspring cohort = 2648 FHS third generation cohort = 1522 Validation cohorts ARICs = 963 GTP = 384	M and F	WB (buffy coat)	Illumina Infinium HumanMethylation450 BeadChip	Differential methylation was observed at 92 CpGs and replicated in the validation cohorts. These CpG sites were associated with cardiovascular disease traits. Some of these CpGs were annotated to genes, such as Lipase A (*LIPA)*, the *ABO blood group* gene (*ABO)*, and *Serologically defined colon cancer antigen 8* (*SDCCAG8)* that showed putative causality for CHD and MI
[59]	Sweden	Monozygotic twin pairs: 83 Dizygotic twin pairs 155 Single twins: 59	M and F	WB	Illumina Infinium HumanMethylation450 BeadChip	Differential methylation at 20 top-ranked CpGs associated with non-stroke CVD, overall stroke, and ischemic stroke. The methylation of these CpGs was shown to determine the levels of cardiometabolic trait (eg., BMI and blood pressure), with the latter mediating CVD risk.
[60]	Turkey	CAD = 1 Control = 1	M	WB	Reduced Representation Bisulfite Sequencing	Eleven genes [*Lipid Droplet Associated Hydrolase* (*LDAH*), *Apolipoprotein B* (*APOB*), *Acyl-CoA synthetase members 2A*, *5 and 3* (*ACSM2A*, *ACSM5*, *ACSM3*), *Carboxylesterase 1* (*CES1*), *Carboxylesterase 1 Pseudogene 1* (*CES1P1*), *AFG3 Like Matrix AAA Peptidase Subunit 2* (*AFG3L2*), *Iron-Sulfur Cluster Assembly Enzyme* (*ISCU*), *SEC14 Like Lipid Binding 2* (*SEC14L2*), and *Microsomal triglyceride transfer protein* (*MTTP*)] involved in fatty acid and cholesterol metabolism were hypomethylated in the healthy twin as compared with twin that presenting with MI.
[61]	Japan	MI = 192 Control = 192	M	WB	Illumina Infinium HumanMethylation450 BeadChip	An association was identified between DNA methylation at 2 CpG sites [cg07786668 in *zinc finer homeobox 3 (ZFHX3)* and cg17218495 in *SWI/SNF-related*, *matrix-associated*, *actin-dependent regulator of chromatin*, *subfamily a*, *member 4* (*SMARCA4*)] and MI, which were independent of CVD risk factors, such as BMI, blood lipid levels, and T2D.
[62]	Germany	Discovery Cohorts: KORA = 1103 NAS = 344 Validation Cohort: InCHIANTI = 443	M and F	PBLs	Illumina Infinium HumanMethylation450 BeadChip	Methylation at 11 CpGs was related to incidents of MI, although after adjusting for medication usage, 9 CpGs remained and composed an epigenetic fingerprint for MI. This included genes such as LDL Receptor Related Protein 8 (*LRP8)* and *KCNN1 potassium calcium-activated channel subfamily N member 1* (*KCCN1)*.

**Table 4 ijms-26-02355-t004:** Classification of main study findings based on disease outcome.

Disease Phenotype	Studies	Main Genes Identified	Summary of Conclusions
MI and CHD	[24,30,32,36,39,41,42,44,50,56,61,62]	*ZBTB12*, *BMPR2*, *IL-6*, *DYSF*, *NRG1*, *AHRR*, *F2RL3*, *MPO*, *ZFHX3*, *FOXP3*, *SOAT1*, *SMARCA4* and *TXNIP*, *CASR*, *GUCAIB*, *HPCAL1*, *PTPRN2*, and *CDH23*	Identified genes involved in pathways related to inflammation, vascular remodeling, and oxidative stress. Identified gene associations independent from traditional risk factors, suggesting an intrinsic epigenetic contribution to disease risk.
Stroke	[45,59]	*ASB10* and *TTC37*	Identified methylation changes may contribute to stroke pathophysiology through mechanisms involving homocysteine metabolism, blood pressure, and BMI regulation. Some stroke-related epigenetic associations were population-specific (e.g., observed only in Europeans).
HF, general CVD, and atherosclerosis	[26,31,37,38,40,43,46,47,48,49,54]	HF severity: *HDAC9*, *JARID2*, *GREM1*, and *SULF1* CVD risk: *CYP8B1*, *SPON1*, *PACS1*, *UACA*, *NPPA*, *AHRR*, and *CPT1A* Atherosclerosis: *IL-6*, *AHRR*, *hTERT*, *GF11*, *LRRC52*, *NR3C1*, *miRNA-233*, *CRISP2*, and *BRCA1*	Hypo- and hypermethylation of distinct CpGs linked to cardiac stress responses, atherosclerosis, and inflammation. Associations with CRP levels and kidney function genes suggest inflammation and metabolic dysfunction as key contributors to CVD risk.
Multiple CVD traits	[27,34,51,52,53,55,58,60]	*ABCG1*, *CPT1A*, *FABP3*, *SREBF1*, *LIPA*, *SDCCAG8*, *ABO*, *TLR2*, *IFN-γ*, and *iNOS*	Identified differentially methylated genes associated with CVD risk factors: lipid metabolism, insulin sensitivity, diastolic blood pressure, and BMI.

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
