# Peer review of "Blood-Based DNA Methylation Biomarkers to Identify Risk and Progression of Cardiovascular Disease"

_ijms, 2025, doi:10.3390/ijms26052355_

Round 1
Reviewer 1 Report
Comments and Suggestions for Authors
The referee can only congratulate the authors on the enormous amount of work they have done. If any doubts remain after reading the discussion, it is not their fault but rather due to the conflicting nature of the existing studies. What the referee can suggest to improve clarity is that the data analysis in the discussion should be conducted based on sex and age, as these might be confounding factors.
Author Response
Reviewer comments:
The referee can only congratulate the authors on the enormous amount of work they have done. If any doubts remain after reading the discussion, it is not their fault but rather due to the conflicting nature of the existing studies. What the referee can suggest to improve clarity is that the data analysis in the discussion should be conducted based on sex and age, as these might be confounding factors.
Authors response:
We value the reviewer's emphasis on the critical issue of study confounders and agree that careful consideration of these variables is paramount for drawing robust conclusions in the discussion. It is indeed well documented that sex and age, as well as other factors including smoking, alcohol consumption, comorbidities and medication use, influence DNA methylation profile of many genes. Unfortunately, of the 46 studies covered in this review, only 33 adjusted for age. Ten studies were conducted exclusively in males, three exclusively in females, while three did not specify the sex of participants. Of the 32 studies which included both males and females, only 23 adjusted for sex. As a result, we are unable to conduct a thorough comparative analysis that properly considers age and sex.
Nonetheless, our review highlighted DNA methylation of five genes, FOXP3, IL-6, CPT1A, ABCG1 and AHRR, which associated with CVD in at least two or more studies. Importantly, these studies accounted for age, sex, and additional covariates such as BMI, smoking, comorbidities, and blood cell count, reinforcing their reliability and potential as CVD biomarkers.
We have now highlighted the inconsistencies in covariate adjustment across the reviewed studies as a limitation in the discussion (page 24, first paragraph). Additionally, we have emphasized the importance of incorporating adjustments for key confounders, such as age and sex, as a standardized practice in future DNA methylation research (page 24, second paragraph).
Reviewer 2 Report
Comments and Suggestions for Authors
This is a narrative review of studies on DNA methylation in blood with cardiovascular disease (CVD) outcomes. The authors screened a total of 4747 articles and identified a total of 46 studies that were of interest. Overall, the authors have done a careful literature review to describe the role of DNA methylation in blood and/or blood cells. The paragraphs are effective in covering the different aspects of the topic, including the methodological approaches and the main limitations.
However, I have some suggestions and points that need to be clarified by the authors.
The main concern is the heterogeneous nature of the phenotypes in the included papers, which did not allow any real conclusion to be drawn. The authors should further define the criteria for the consideration of studies for this review (e.g. types of studies, types of participants, types of outcomes) as well as the period of publication.
Furthermore, the authors should better comment on the fact that the different methods for measuring DNA methylation and the variety of clinical phenotypes limit comparisons and prevent a deeper synthesis of existing knowledge.
Author Response
Reviewer comments:
This is a narrative review of studies on DNA methylation in blood with cardiovascular disease (CVD) outcomes. The authors screened a total of 4747 articles and identified a total of 46 studies that were of interest. Overall, the authors have done a careful literature review to describe the role of DNA methylation in blood and/or blood cells. The paragraphs are effective in covering the different aspects of the topic, including the methodological approaches and the main limitations.
However, I have some suggestions and points that need to be clarified by the authors.
Comment 1: The main concern is the heterogeneous nature of the phenotypes in the included papers, which did not allow any real conclusion to be drawn.
Author Response: As cardiovascular disease is an umbrella term encompassing a wide range of phenotypes, our review included studies that examined both general CVD outcomes as well as related conditions affecting the heart and blood vessels. To this end, we included studies investigating myocardial infarction, coronary heart disease, stroke, heart failure, and atherosclerosis, recognizing the shared pathophysiological mechanisms and overlapping risk factors of these conditions. To better reflect this, we have now provided a clearer description of the inclusion criteria on page 2, paragraph 4. Additionally, to enhance the interpretation of our findings, we have included a supplementary table summarizing the identified gene associations and the conclusions drawn after stratifying the studies by condition outcomes.
Comment 2: The authors should further define the criteria for the consideration of studies for this review (e.g. types of studies, types of participants, types of outcomes) as well as the period of publication.
Author Response: This review included randomized controlled trials, cohort studies, case-control studies, cross-sectional, and longitudinal studies conducted in humans with CVD or related conditions affecting the heart and blood vessels, including myocardial infarction, coronary heart disease, stroke, heart failure, and atherosclerosis. To maximize the sample size of included articles, no date restrictions were applied in the literature search. we have now provided a clearer description of the inclusion criteria on page 2, paragraph 4.
Comment 3: Furthermore, the authors should better comment on the fact that the different methods for measuring DNA methylation and the variety of clinical phenotypes limit comparisons and prevent a deeper synthesis of existing knowledge.
Author Response: We thank the reviewer for highlighting this issue. We have added a statement in the discussion addressing the range of CVD phenotypes and methodological approaches included in our review, which present challenges for comparative analysis (page 24, first paragraph).
Round 2
Reviewer 2 Report
Comments and Suggestions for Authors
I suggest that the supplementary table summarising the gene associations identified and the conclusions drawn after stratification of the studies by disease outcome be included in the paper
Author Response
Reviewer comment: I suggest that the supplementary table summarising the gene associations identified and the conclusions drawn after stratification of the studies by disease outcome be included in the paper.
Author response: We thank the reviewer for this suggestion. The supplementary table has now been included in the main article as Table 4 and referred to in the discussion.